# SGLT2 Inhibitors: The Star in the Treatment of Type 2 Diabetes?

**DOI:** 10.3390/diseases8020014

**Published:** 2020-05-11

**Authors:** Yoshifumi Saisho

**Affiliations:** Department of Internal Medicine, Keio University School of Medicine, Tokyo 1608582, Japan; ysaisho@keio.jp; Tel.: +81-3-3353-1211

**Keywords:** sodium-glucose cotransporter 2 inhibitor, type 2 diabetes, cardiovascular outcome trial, cardiorenal protection

## Abstract

Sodium-glucose cotransporter 2 (SGLT2) inhibitors are a novel class of oral hypoglycemic agents which increase urinary glucose excretion by suppressing glucose reabsorption at the proximal tubule in the kidney. SGLT2 inhibitors lower glycated hemoglobin (HbA1c) by 0.6–0.8% (6–8 mmol/mol) without increasing the risk of hypoglycemia and induce weight loss and improve various metabolic parameters including blood pressure, lipid profile and hyperuricemia. Recent cardiovascular (CV) outcome trials have shown the improvement of CV and renal outcomes by treatment with the SGLT2 inhibitors, empagliflozin, canagliflozin, and dapagliflozin. The mechanisms by which SGLT2 inhibitors improve CV outcome appear not to be glucose-lowering or anti-atherosclerotic effects, but rather hemodynamic effects through osmotic diuresis and natriuresis. Generally, SGLT2 inhibitors are well-tolerated, but their adverse effects include genitourinary tract infection and dehydration. Euglycemic diabetic ketoacidosis is a rare but severe adverse event for which patients under SGLT2 inhibitor treatment should be carefully monitored. The possibility of an increase in risk of lower-extremity amputation and bone fracture has also been reported with canagliflozin. Clinical trials and real-world data have suggested that SGLT2 inhibitors improve CV and renal outcomes and mortality in patients with type 2 diabetes (T2DM), especially in those with prior CV events, heart failure, or chronic kidney disease. Results of recent trials including individuals without diabetes may change the positioning of this drug as ″a drug for cardiorenal protection″. This review summarizes the potential of SGLT2 inhibitors and discusses their role in the treatment of T2DM.

## 1. Introduction

Sodium-glucose cotransporter 2 (SGLT2) inhibitors are a novel class of oral hypoglycemic agents (OHA). They have been developed based on the discovery of phlorizin, a natural product with SGLT inhibitory activity which was extracted from the bark of the apple tree in 1835, and advances in understanding of the mechanisms of glucose transport, resulting in the identification of SGLTs and their functional properties in the 1980–1990s [1]. In Japan, the first SGLT2 inhibitor, ipragliflozin, was marketed in 2014, and currently six SGLT2 inhibitors, ipragliflozin, dapagliflozin, canagliflozin, empagliflozin, luseogliflozin, and tofogliflozin, are available for the treatment of type 2 diabetes (T2DM) (Table 1). Ipragliflozin and dapagliflozin have also been approved for the treatment of type 1 diabetes (T1DM) in Japan.

SGLT2 inhibitors improve hyperglycemia, reduce body weight and visceral adiposity, and improve various metabolic abnormalities associated with metabolic syndrome such as blood pressure, lipid profile, and serum uric acid level [3]. Further, recent cardiovascular outcome trials (CVOTs) have shown improvement of CV and renal outcomes in patients with and even without T2DM by treatment with SGLT2 inhibitors, and the American Diabetes Association/the European Association of the Study for Diabetes (EASD) recommend SGLT2 inhibitors as the mainstay of treatment of T2DM [4].

In this minireview, the potential of SGLT2 inhibitors is summarized and current and future perspectives of this drug class in the treatment of T2DM are discussed.

## 2. Mechanisms of Action of SGLT2 Inhibitors

In the kidney, approximately 180 g of glucose per day is excreted in the primitive urine through glomerular filtration. Most of the glucose in the primitive urine is however completely reabsorbed by SGLT2 and SGLT1 expressed in the proximal tubule, and glucose is not normally excreted in the urine.

SGLT activity mediates apical sodium and glucose transport across cell membranes. Cotransport is driven by active sodium extrusion by the basolateral sodium/potassium-ATPase, thus facilitating glucose uptake against an intracellular up-hill gradient. Basolaterally, glucose exits the cell through facilitative glucose transporter 2 (GLUT2). In humans, six SGLT isoforms have been identified [1,5].

SGLT2 is responsible for glucose reabsorption in the proximal tubule segment 1 and 2 (S1/2), wherein it reabsorbs more than 90% of the filtered glucose load, while normally SGLT1 reabsorbs the residual glucose in the proximal tubule segment 3 (S3). However, in SGLT2 knockout mice, SGLT1 compensated and reabsorbed up to 35% of the filtered glucose load [1,5]. Glucose resorption by SGLT2 is increased by 30% in the setting of hyperglycemia [6], although it remains unclear whether SGLT2 expression is increased in patients with diabetes [7].

SGLT2 inhibitors suppress reabsorption of glucose by inhibition of SGLT2, and thereby increase urinary glucose excretion by approximately 60–80 g per day and ameliorate hyperglycemia [3]. Excretion of 60–80 g of excess glucose corresponds to 240–320 kcal of energy loss from the body, promoting weight loss. Improvement of obesity/overweight, especially abdominal fat accumulation promotes amelioration of insulin resistance and results in improvement of metabolic parameters such as blood pressure, lipid profile, and serum uric acid level (Figure 1).

As SGLT1 is responsible for glucose absorption in the small intestine, SGLT2 inhibitors with low SGLT1/SGLT2 selectivity such as canagliflozin have been shown to suppress postprandial glucose excursion and increase glucagon-like peptide 1 (GLP-1) secretion in addition to increasing urinary glucose excretion [8,9].

On the other hand, given the mechanisms of action of SGLT2 inhibitors, their glucose-lowering effect is attenuated with reduced renal function. Thus, treatment with SGLT2 inhibitors is not recommended in patients with renal dysfunction, i.e., estimated glomerular filtration rate (eGFR) <45 mL/min, from the glucose-lowering point of view. However, in view of the results of CVOTs showing a renoprotective effect of SGLT2 inhibitors among those with a wide range of renal function [10,11], currently treatment with SGLT2 inhibitors is rather also considered for patients with diabetes and renal dysfunction, i.e., eGFR ≥ 30 mL/min, to improve renal outcomes [4]. Since glucose reabsorption by SGLT1 is compensatorily enhanced as a result of SGLT2 inhibition [2,5], SGLT2 inhibitor monotherapy does not increase the risk of hypoglycemia [12].

## 3. Clinical Efficacy of SGLT2 Inhibitors

### 3.1. Glucose-Lowering Effect

A meta-analysis of 45 clinical trials showed that treatment with SGLT2 inhibitors results in a HbA1c reduction of 0.79% with monotherapy and 0.61% with add-on therapy to other glucose-lowering agents in patients with T2DM [13]. SGLT2 inhibitors improve both fasting and postprandial hyperglycemia and increase time-in-range (TIR, proportion of time spent in the target glucose range between 70 and 180 mg/dL) assessed by continuous glucose monitoring (CGM) [14,15].

### 3.2. Body Weight, Blood Pressure, and Other Metablic Parameters

The above-mentioned meta-analysis showed that treatment with SGLT2 inhibitors in patients with T2DM reduced body weight by 1.7 kg (2.4%), and systolic and diastolic blood pressure by 4 and 2 mmHg, respectively, without increasing heart rate. Reduction in serum triacylglycerol level by 1–9% and serum uric acid level by 0.3–0.9 mg/dL, and increase in serum HDL-cholesterol by 6–9% by treatment with SGLT2 inhibitors have also been reported in patients with T2DM [13].

On the other hand, increase in LDL-cholesterol by 2–6% by treatment with SGLT2 inhibitors has been reported. However, Hayashi et al. have reported that small dense LDL-cholesterol, a more atherogenic subspecies of LDL-cholesterol, is reduced while less atherogenic large buoyant LDL-cholesterol is increased by 12-week treatment with dapagliflozin [16].

### 3.3. Hypoglycemia

The risk of hypoglycemia with SGLT2 inhibitors is low, and the risk of hypoglycemia is similar to placebo when SGLT2 inhibitors are used as monotherapy [12,13]. However, the risk of hypoglycemia may be increased when SGLT2 inhibitors are used in combination with insulin and/or insulin secretagogues. Therefore, the dose of insulin and/or insulin secretagogues may need to be reduced when combined with an SGLT2 inhibitor, to avoid hypoglycemia. When reducing the dosage of insulin, adjustment by within 10–20% of the total insulin dose is recommended to avoid development of diabetic ketoacidosis [17].

### 3.4. Beta Cell Function

Since SGLT2 inhibitors lower plasma glucose level independently of insulin, they do not increase insulin secretion but rather improve beta cell function [18,19] through amelioration of glucotoxicity and possibly reduction of beta cell workload [20,21,22]. On the other hand, glucagon secretion increases after administration of SGLT2 inhibitors, possibly because of rapid loss of glucose from the body [23,24]. Elevated plasma glucagon level also contributes to promoting lipolysis and reducing liver fat and visceral adiposity [25]. A direct effect of SGLT2 inhibitors on alpha cells has also been proposed [26,27], though there are conflicting results [28,29] and further research is warranted.

## 4. EMPA-REG OUTCOME Trial

Since SGLT2 inhibitors showed improvement of multiple CV risk factors in addition to hyperglycemia in preclinical studies, treatment with SGLT2 inhibitors had been expected to reduce the risk of CV events. Nonetheless, the marked reduction in 3-point major adverse cardiovascular events (MACE) shown in the EMPA-REG OUTCOME (Empagliflozin Cardiovascular Outcome Event Trial in Type 2 Diabetes Mellitus Patients-Removing Excess Glucose) trial [30] even exceeded the expectation.

### 4.1. Cardiovascular Outcome

The EMPA-REG OUTCOME trial was a randomized controlled trail to assess CV safety and efficacy of empagliflozin 10 mg and 25 mg daily compared with placebo, added onto standard therapy [30]. A total of 7020 patients with T2DM and prior CV events were enrolled and followed up for a median of 3.1 years. Treatment with empagliflozin resulted in significant reduction in the primary endpoint (3-point MACE; a composite of CV death, non-fatal myocardial infarction and non-fatal stroke) by 14% and especially CV death by 38%. Actuarial analysis using data from the EMPA-REG OUTCOME trial estimated that empagliflozin improves survival by 1 to 5 years in patients with T2DM and established CV disease (CVD) [31].

According to the results of UKPDS (UK Prospective Diabetes Study) [32], it has been assumed that the effect of glucose-lowering therapy to improve CV outcome appears after 5 to 10 years of treatment. However, surprisingly, the reduction in CV events by empagliflozin was already observed only a few months after starting treatment [30]. Further, the incidence of hospitalization for heart failure was significantly reduced by 35% by empagliflozin treatment, indicating that the improvement of CV outcome by empagliflozin treatment was likely due to its hemodynamic effect rather than anti-atherosclerotic effect.

Although various hypotheses have been proposed as mechanisms of the improvement of CV outcomes by treatment with SGLT2 inhibitors [3,33], osmotic diuresis due to glucosuria likely plays a major role that reduces cardiac preload and thereby reducing the incidence of heart failure and arrythmia, resulting in the reduction in CV death (Figure 1). It has been reported that treatment with dapagliflozin 10 mg reduced plasma volume by 9% [34]. Increase in blood ketone bodies and hematocrit may also contribute to cardioprotection by treatment with SGLT2 inhibitors.

### 4.2. Renal Outcome

In addition to the improvement of CV outcome, renal events (a composite of progression to macroalbumiuria, doubling of serum creatinine level, initiation of renal-replacement therapy, or death from renal disease) were significantly reduced by 39% by treatment with empagliflozin in the EMPA-REG OUTCOME study [35]. The mechanism by which empagliflozin improves renal outcome is likely through improvement of glomerular hyperfiltration [36]. Empagliflozin promotes natriuresis in the proximal tubule, which results in reducing glomerular hyperfiltration through tubuloglomerular feedback. Renal protection by empagliflozin may also promote erythropoietin production by the kidney [37,38], contributing to an increase in hematocrit (Figure 1).

Subsequently, a second CVOT with an SGLT2 inhibitor, the CANVAS/CANVAS-R (Canagliflozin Cardiovascular Assessment Study) trial, has been reported in 2017 [39]. A total of 10,142 patients with T2DM and at high risk of CVD were enrolled and randomly assigned to treatment with either canagliflozin 100–300 mg daily or placebo added onto standard therapy, and followed up for a median of 3.1 years. This study also showed a reduction in 3-point MACE by 14%, hospitalization for heart failure by 33% and renal events by 40% by treatment with canagliflozin compared with placebo, suggesting that cardiorenal protection is a class effect of SGLT2 inhibitors (Table 2). Thirty-percent of the participants in the CANVAS/CANVAS-R trial had no prior CV events [40]. A meta-analysis of three CVOTs with SGLT2 inhibitors has confirmed the improvement of CV outcomes in those with T2DM and prior CV events but not in those with T2DM without prior CV events [41]. However, an observational study using real-world data of more than 300,000 patients with T2DM, the majority of whom were without established CVD, also showed that the use of SGLT2 inhibitors resulted in a reduction in hospitalization for heart failure by 39% and all-cause mortality by 51% [42]. A renoprotective effect of treatment with canagliflozin has also been shown in patients with T2DM and chronic kidney disease (CKD, mean eGFR 56.2 mL/min, median urinary albumin [mg] to creatinine [g] ratio [ACR] 927) in the CREDENCE (Canagliflozin and Renal Endpoints in Diabetes with Established Nephropathy Clinical Evaluation) trial [43], and a more recent meta-analysis including this trial has suggested that treatment with SGLT2 inhibitors improves CV outcome irrespective of established CV history or kidney function among patients with T2DM [44]. A list of published CVOTs with SGLT2 inhibitors is shown in Table 3.

## 5. Adverse Effects

The adverse effects of SGLT2 inhibitors other than hypoglycemia include genitourinary tract infection and dehydration and related symptoms [3,12,13]. Genital infection more frequently occurs in female patients. Cases of Fournier′s gangrene associated with the use of SGLT2 inhibitors have also been reported [47]. Body weight loss induced by SGLT2 inhibitors may increase the risk of development of sarcopenia in elderly patients. Cases of diabetic ketoacidosis after the initiation of SGLT2 inhibitors have been reported [17]. Ketoacidosis can develop without hyperglycemia, i.e., euglycemic diabetic ketoacidosis. Patients should be carefully monitored for the development of ketoacidosis, especially those with T1DM and those with T2DM and insulin deficiency [17]. In the CANVAS/CANVAS-R trial, increased risk of lower-extremity amputation and bone fracture was observed in the canagliflozin group [39], although there was no significant difference in the rate of amputation or fracture in the CREDENCE trial [43].

## 6. Positioning of SGLT2 Inhibitors in Treatment of T2DM

Obesity is an established risk factor for the development of T2DM. SGLT2 inhibitors not only improve hyperglycemia but also induce weight loss, ameliorating the pathogenesis of T2DM. Although lifestyle modification is important for weight loss in the treatment of T2DM, it is often difficult to maintain long-term weight loss.

It can be assumed that the same effect would be obtained if patients restricted carbohydrate intake by the same amount as that excreted by SGLT2 inhibitors, i.e., 60–80 g per day (240–320 kcal). However, intensive lifestyle modification failed to improve CV outcome in the Look AHEAD (Action for Health in Diabetes) trail [48]. The impressive results observed in CVOTs with SGLT2 inhibitors [30,39,43,45] suggest that SGLT2 inhibitors promote cardiorenal protection through specific effects such as diuretic effects, apart from the effects of caloric restriction.

The ADA/EASD now positions SGLT2 inhibitors as the mainstay in the management of T2DM [4]. The use of SGLT2 inhibitors is recommended for patients with a high risk of or established atherosclerotic cardiovascular disease (ASCVD) and those with CKD or heart failure.

The target population of SGLT2 inhibitors is being expanded also in Japan. At the time of launch of the first SGLT2 inhibitor in Japan, ipragliflozin, in 2014, the target population of SGLT2 inhibitors was thought to be rather restricted to obese, younger patients with T2DM and metabolic syndrome (Table 4), while about half of patients with T2DM are not obese in Japan. However, currently, treatment with SGLT2 inhibitors is also considered for patients with T2DM, and especially those with established ASCVD, heart failure, or CKD, as recommended by the ADA/EASD. Furthermore, improvement of CV outcome by treatment with dapagliflozin has been observed in patients with heart failure with reduced ejection fraction (HFrEF), irrespective of the presence or absence of diabetes [46,49], suggesting the possibility that SGLT2 inhibitors can be used as a drug for heart failure independent of the presence or absence of diabetes. Trials evaluating the efficacy of SGLT2 inhibitors in patients with heart failure with preserved ejection fraction (HFpEF) and those with CKD and without diabetes are also under way [50,51]. Since SGLT2 inhibitors lower plasma glucose level independently of insulin, they can also be used for patients with T1DM as concomitant medication with insulin [52].

Table 5 summarizes the points required for antidiabetic medications. SGLT2 inhibitors fulfil many of these points, indicating their great potential in the management of T2DM.

Meta-analysis suggests that the cardiorenal benefits of SGLT2 inhibitors are a class effect; however, the structure, dosage, pharmacokinetic/pharmacodynamic (PK/PD) profile, and SGLT2 selectivity are different among SGLT2 inhibitors. Further research is needed to clarify whether there is any difference in the effect on clinical outcomes among different SGLT2 inhibitors. The first dual SGLT1/2 inhibitor, sotagliflozin, has also been developed, and a CVOT with sotagliflozin is ongoing [53,54].

## 7. Conclusions

SGLT2 inhibitors are a novel class of anti-diabetic medication that lowers plasma glucose level without hypoglycemia by increasing urinary glucose excretion. SGLT2 inhibitors also reduce body weight and abdominal fat, reduce blood pressure, and improve lipid profile and serum uric acid level. Considering the improvement of CV outcome, i.e., the goal of treatment of T2DM, by treatment with SGLT2 inhibitors shown in CVOTs, SGLT2 inhibitors are no longer considered to be oral hypoglycemic agents but rather medication for diabetic complications or for cardiorenal protection. The expected potential of cardiorenal protection by SGLT2 inhibitors in the non-diabetic population further strengthens this paradigm shift in the concept of SGLT2 inhibitors.

On the other hand, although SGLT2 inhibitors are generally well tolerated, their adverse effects including genitourinary tract infection, dehydration, and ketoacidosis should be carefully monitored. The initiation of SGLT2 inhibitors should be judged in each individual patient according to the balance between benefits and drawbacks (Figure 2).

Given the cardiorenal benefits of treatment with SGLT2 inhibitors, they can be considered the star or blockbuster for the treatment of T2DM. Despite this, weight loss induced by SGLT2 inhibitors reaches a plateau in the long term, which may be due to an increase in food intake with increased appetite [55]. Thus, the importance of continuous lifestyle modification should be emphasized to patients when starting SGLT2 inhibitors, as with other anti-diabetic medications. Although SGLT2 inhibitors provide cardiorenal benefits independently of glucose- and weight-reducing effects, lifestyle modification and weight loss remain the most important and fundamental therapy for patients with T2DM [20,21,22]. SGLT2 inhibitors should be used to support weight loss induced by lifestyle modification, which will also motivate patients to have a healthy lifestyle. In this regard, combination of SGLT2 inhibitor with metformin and/or a GLP-1 receptor agonist may foster further weight loss without increasing the risk of hypoglycemia. SGLT2 inhibitors can be the star in the treatment of T2DM, when used appropriately through a patient-centered approach.

## Figures and Tables

**Figure 1 diseases-08-00014-f001:**
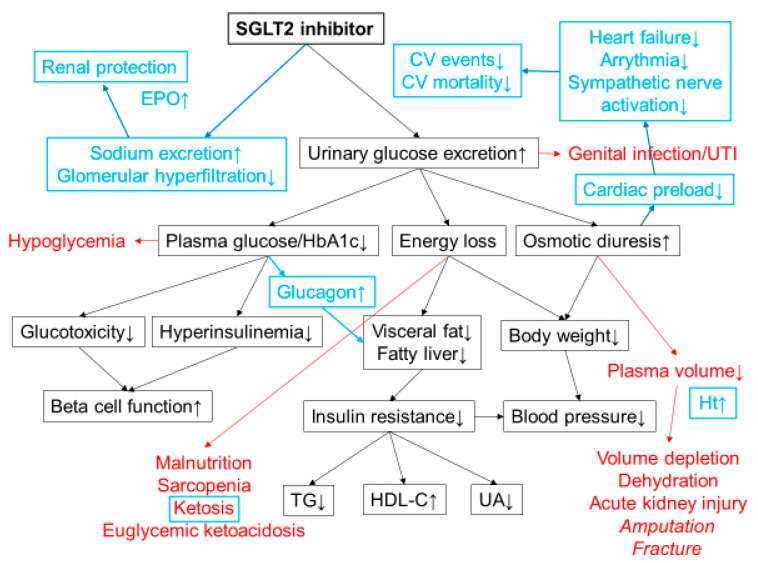
Mechanisms of action of SGLT2 inhibitors. SGLT2 inhibitors lower plasma glucose level by increasing urinary glucose excretion. Energy loss by SGLT2 inhibitor treatment promotes weight loss and improves insulin resistance and various metabolic parameters. Possible adverse effects are shown in red. Increased risk of lower-extremity amputation and bone fracture has been reported in a clinical trial with canagliflozin. The cardiorenal benefits shown in CVOTs have revealed additional mechanisms of action of SGLT2 inhibitors that were unknown at the time of launch (highlighted in blue). Osmotic diuresis and natriuresis are likely to be the major mechanisms of the cardiorenal benefits of SGLT2 inhibitors. Increases in blood ketone bodies and hematocrit may also contribute to the cardiorenal benefits. CV; cardiovascular, TG; triacylglycerol, HDL-C; high density lipoprotein cholesterol, UA; uric acid, Ht; hematocrit, UTI; urinary tract infection, EPO; erythropoietin.

**Figure 2 diseases-08-00014-f002:**
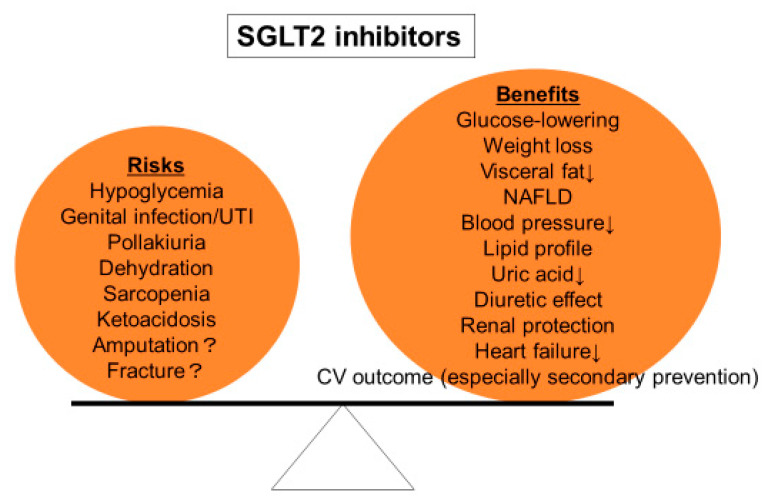
Risks and benefits of SGLT2 inhibitors. Treatment with SGLT2 inhibitors should be decided according to the balance between risks and benefits in each individual patient.

**Table 1 diseases-08-00014-t001:** Sodium-glucose cotransporter 2 (SGLT2) inhibitors available in Japan.

Generic Name	Dosage	SGLT1/2 Selectivity	Half-Life (t_1/2_)	Indication
Ipragliflozin	50–100 mg once daily	254:1	15 h	Type 1 and type 2 diabetes
Dapagliflozin	5–10 mg once daily	1242:1	8–12 h	Type 1 and type 2 diabetes
Canagliflozin	100 mg once daily	155:1	12 h	Type 2 diabetes
Empagliflozin	10–25 mg once daily	2680:1	14–18 h	Type 2 diabetes
Luseogliflozin	2.5–5.0 mg once daily	1770:1	9 h	Type 2 diabetes
Tofogliflozin	20 mg once daily	2912:1	5 h	Type 2 diabetes

SGLT1/2 selectivity: ratio of IC_50_ for SGLT1 to IC_50_ for SGLT2 (fold). Adapted from reference [2].

**Table 2 diseases-08-00014-t002:** Effects on CVD risk among glucose-lowering agents.

Specific Effects on CVD Risk	Non-Specific Effects on CVD Risk
Metformin?	DPP-4 inhibitors
Pioglitazone	Sulfonylureas
GLP-1 receptor agonists *	Glinide
SGLT2 inhibitors *	Alpha-glucosidase inhibitors? Insulin

* evidenced by CVOTs.

**Table 3 diseases-08-00014-t003:** Cardiovascular outcome trials (CVOTs) with SGLT2 inhibitors.

SGLT2 Inhibitor	Trial Name	Publication Year	Reference
Empagliflozin	EMPA-REG OUTCOME	2015	[30]
Canagliflozin	CANVAS/CANVAS-R	2017	[39]
	CREDENCE	2019	[43]
Dapagliflozin	DECLARE-TIMI 58	2019	[45]
	DAPA-HF	2019	[46]

DECLARE-TIMI; dapagliflozin effect on cardiovascular events-thrombolysis in myocardial infarction, DAPA-HF; dapagliflozin and prevention of adverse-outcomes in heart failure.

**Table 4 diseases-08-00014-t004:** Patients considered for SGLT2 inhibitor treatment in the past, the present, and the future.

Patients Considered for SGLT2 Inhibitor Treatment
At time of launch:Type 2 diabetes (T2DM)Particularly, young to middle-aged patients with obesity (metabolic syndrome)

Current: in addition to the above patients with T2DM:Those with CVDThose with heart failure, especially with reduced ejection fraction (HFrEF)Those with diabetic kidney disease (DKD) and eGFR ≥ 30 mL/minThose with type 1 diabetes

Future: in addition to the above:Those with HFrEF, without diabetesThose with heart failure with preserved ejection fraction (HFpEF)Those with chronic kidney disease (CKD) without diabetes?Those with prediabetes (for prevention of T2DM)?

**Table 5 diseases-08-00014-t005:** Potential of SGLT2 inhibitors in treatment of T2DM.

Points Required for Antidiabetic Medication	SGLT2 Inhibitors
Glucose (HbA1c)-lowering efficacy	Moderate
Low risk of hypoglycemia	Yes
Reduction of body weight	Yes
Improvement of postprandial glycemic excursion	Yes, but not specifically
Improvement of (peripheral) hyperinsulinemia	Yes
Tolerability/adverse events	Yes, but there are drug-specific adverse effects
Improvement of beta cell function/beta cell protection	Yes
Prevention of microvascular complications	Yes (especially nephropathy)
Prevention of CVD	Yes (especially secondary prevention)
Low cost	No
Long-term safety	Yes?
Extension of healthy longevity/QALY	Yes?

QALY; quality-adjusted life-years.

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
