# Peer review of "SGLT2 Inhibitors: The Star in the Treatment of Type 2 Diabetes?"

_diseases, 2020, doi:10.3390/diseases8020014_

Round 1

Reviewer 1 Report

This review manuscript by Saisho provided the summary of recent clinical trials using SGLT2 inhibitors in humans. It also discussed the pros and cons in treatment of SGLT2 inhibitors. Overall, this review was written in the well-organized structure and included the constructive point of views in the therapeutic potential and considerations of clinical treatment of SGLT2 inhibitors. Here are some suggestions for this manuscript.

Major suggestions

  1. Since the author discussed that using different type of inhibitors in different trials led to different benefits and risks. That would be better to create a table to summarize the beneficial outcome and adverse effect in different SGLT2 inhibitors treatment. Like the following suggestion,

Type of SGLT2

Trial name

Beneficial effects

Adverse effects

References

empagliflozin

EMPA-REG OUTCOME

canagliflozin

CANVAS/CANVAS-R

reduction in 3-point MACE…

bone fracture..

  1. It seems like different SGLT2 inhibitors has different outcomes in different trials. It would be better to compare and discuss what is the underlying mechanisms causing the different outcomes between these inhibitors (structure, specificity, dosage, inhibitory efficacy of SGLT2).

Minor suggestions

  • Provide reference for the description in Line 68-70 at Page 2, “the glucose-lowering 68 effect of SGLT2 inhibitors is attenuated with reduced renal function. Thus, treatment with SGLT2 69 inhibitors is not recommended in patients with severe renal dysfunction, i.e., eGFR <30 ml/min”
  • Provide reference for the description in Line 71-72 at Page 2, “glucose reabsorption by SGLT1 is compensatorily enhanced as a result of SGLT2 inhibition”.
  • Provide a full name of abbreviations when the author mentioned them at first time.

Line 101, EMPA-REG OUTCOME

Line 113, UKPDS

Line 134, CANVAS/CANVAS-R

Line 148, CREDENCE

Line 171, AHEAD

Reviewer 2 Report

Yoshifumi Saisho submits a review entitled "SGLT2 inhibitors: The star in the treatment of type 2 diabetes?".

In this review the author summarizes the potential of SGLT2 inhibitors and discusses their role in the treatment of T2DM. This review is clearly and nicely written, but reads a bit too much as an advertisement for these new drugs. As a matter of perspective some description/consideration should be given to other drug treatments of T2DM and to lifestyle modification.

It would be nice to have more information/background on structure/fuction of Sodium-glucose cotransporter 2. Any information on Knock-Out pre clinical models?

The sentence "Thus, treatment with SGLT2 69 inhibitors is not recommended in patients with severe renal dysfunction, i.e., eGFR <30 ml/min." needs a reference.

Minor:

in abstract the developed term for T2DM is not defined, same for HbA1c.

lines 87-88 "small dense LDL-cholesterol, which is more atherogenic among LDL-cholesterol," should be rephrased into a clearer sentence.

line 140 "The CANVAS/CANVAS-R trial contained patients 140 without prior CV events at 30%" does that mean 30% of patients?
